# Association between myopia progression and quantity of laser treatment for retinopathy of prematurity

Eileen S. Hwang [ORCID][1]*, Iris S. Kassem [ORCID][2], Rawan Allozi[3], Sasha Kravets[3], Khalid Y. Al-Kirwi [ORCID][2], Joelle A. Hallak[4], Deborah M. Costakos[2]

1 Department of Ophthalmology and Visual Sciences, University of Utah, Salt Lake City, Utah, United States of America, 2 The Eye Institute, Department of Ophthalmology and Visual Sciences, Medical College of Wisconsin, Milwaukee, Wisconsin, United States of America, 3 Department of Epidemiology and Biostatistics, School of Public Health, University of Illinois at Chicago, Chicago, Illinois, United States of America, 4 Department of Ophthalmology and Visual Sciences, University of Illinois at Chicago, Chicago, Illinois, United States of America

* eileenhwang@yahoo.com

## Abstract

**Data Availability Statement:** All relevant data are within the paper and its Supporting Information files.

### Background

Previous studies found that infants with retinopathy of prematurity (ROP) who were treated for more posterior disease with a greater number of laser spots developed higher myopia. These studies included multiple physicians with variations in laser density. In treatments by a single physician, laser spot count is a better surrogate for area of avascular retina and anterior-posterior location of disease, so that the relationship with myopia can be better assessed.

### Methods

Our retrospective study included infants treated with laser for ROP by a single surgeon at a single center. Exclusion criteria were irregularities during laser and additional treatment for ROP. We assessed correlation between laser spot count and change in refractive error over time using a linear mixed effects model.

### Results

We studied 153 eyes from 78 subjects treated with laser for ROP. The average gestational age at birth was 25.3±1.8 weeks, birth weight 737±248 grams, laser spot count 1793±728, and post-treatment follow up 37±29 months. Between corrected ages 0–1 years, the mean spherical equivalent was +0.4±2.3 diopters; between ages 1–2, it was -1.3±3.2D; and ages 2–3 was -0.8±3.1D. Eyes that received more laser spots had significantly greater change in refractive error over time (0.30D more myopia per year per 1000 spots). None of the eyes with hyperopia before 18 months developed myopia during the follow-up period.

**Funding:** This study was supported by an unrestricted grant from Research to Prevent Blindness https://www.rpbusa.org/rpb/ (University of Utah), the Alsam Foundation https://fconline. foundationcenter.org/fdo-grantmaker-profile/?key= ALSA001 (University of Utah), the National Institutes of Health EY014800 (University of Utah) and K08 EY024645 (ISK), and the Children's Research Institute https://childrenswi.org/medical-professionals/research/about-childrens-research-institute (ISK). This investigation was conducted in part in a facility constructed with support from a Research Facilities Improvement Program, grant number C06RR016511 from the National Center for Research Resources, National Institutes of Health (Medical College of Wisconsin). Its contents are solely the responsibility of the authors and do not necessarily represent the official views of the National Institutes of Health. The funders had no role in study design, data collection and analysis, decision to publish, or preparation of the manuscript.

**Competing interests:** I have read the journal's policy and the authors of this manuscript have the following competing interests: Eileen S. Hwang received gifts, meals, travel support and/or education from Regeneron, Allergan, Valeant, Alimera Sciences, Alcon, Bausch & Lomb, Beaver Vistec International, Spark Therapeutics, and Katalys. Iris S. Kassem's spouse is employed by AbbVie. Iris S. Kassem became employed by Novartis after completing work on this project. Rawan Allozsi was employed by AbbVie prior to commencing work on this project and became employed at The Janssen Pharmaceutical Companies of Johnson & Johnson after completing work on this project. This does not alter our adherence to PLOS ONE policies on sharing data and materials.

## Conclusions

Greater myopia developed over time in infants with ROP treated by laser to a larger area of avascular retina.

## Introduction

Untreated retinopathy of prematurity (ROP) can lead to visually devastating outcomes such as retinal detachment. Fortunately, treatment can prevent preterm infants from developing severe vision loss [1,2]. Laser treatment prevents vision loss from macular dragging, retinal detachment, and vitreous hemorrhage. However, even when infants are successfully treated for ROP, they are at a lifelong risk of developing additional vision-threatening problems such as cataract, glaucoma, strabismus, amblyopia, retinal detachment, and refractive error, including myopia.

Myopia occurs after ROP treatment, and the risk of myopia is also elevated in spontaneously regressed ROP [3,4]. The risk of myopia is higher after spontaneous regression of posterior disease (i.e. zone 2) compared to anterior disease (i.e. zone 3) [5,6]. The CRYO-ROP study found more high myopia in treated compared to control eyes, although there were more eyes that could not be refracted in the control group [7]. In the subgroup of subjects that could be refracted in both eyes, there was no difference in myopia rates between treated and control eyes [7]. Anterior-posterior location of active ROP disease correlates with myopia, as well as the extent of final vascularization after spontaneous regression [5]. During laser treatment for ROP, the entire area of avascular retina is ablated to reduce the drive for neovascularization. Posterior location of ROP, and thereby a larger area of avascular retina ablated during laser treatment, may correlate with a greater risk of myopia [8–12]. Precisely measuring the anterior-posterior location or the area of retina treated are difficult. Laser spot count may serve as a surrogate, and previous studies have correlated laser spot counts and zone of treatment with myopia [8–12]. However, these studies included data from multiple treating physicians, and therefore, the number of laser spots could have reflected variations in treatment density between physicians rather than the area of retina treated. We sought to evaluate correlation between myopia and laser spot count after near-confluent treatment by a single ophthalmologist to eliminate variations in spot density as a confounder.

## Methods

The Children's Hospital of Wisconsin Institutional Review Board approval was obtained with a waiver of informed consent. This study conformed to the requirements of the United States Health Insurance Portability and Privacy Act. A retrospective chart review was conducted to screen 164 subjects treated for ROP from January 1, 2010 to July 1, 2019.

The primary outcome measure was refractive error as spherical equivalent, measured by cycloplegic refraction with cyclopentolate. Only subjects treated a single time with laser photocoagulation by a single provider (DMC) whose charts contained complete information about laser spot count and refractive error with cycloplegia from at least one clinic visit were included in the primary analysis. Laser photocoagulation was performed with a 810 nm indirect laser with a 28 D lens to apply near-confluent spots (less than 1/4 spot width apart) to the entire avascular retina between the ora serrata to the vascular-avascular border. The minimum power used for a treatment was 191 ± 25 mW (mean ± SD) and the maximum power used was

217 ± 54 mW. We excluded 6 eyes with equipment difficulties during laser as this may have changed the laser spot count. 7 eyes that subsequently underwent intraocular surgery were also excluded since it may have altered emmetropization in these eyes. We excluded 63 eyes that received treatment with intravitreal injections at any time and 4 eyes that underwent more than one laser treatment as the effect of anti-VEGF agents and multiple lasers may confound results.

## Statistical analysis

The refractive error spherical equivalent between ages 0–1, 1–2, and 2–3 years were calculated by using the latest cycloplegic retinoscopy measured within that time period. Spherical equivalent and laser spots were treated as continuous variables. Mean ± Standard Deviation (SD) were reported for continuous variables. Bivariate and multivariate linear mixed models with random subject intercept, random intercept for eye within subject, and random eye slope were used to evaluate the association between the number of laser spots used in treatment and change in spherical equivalent over time. A mixed effects model was used for final analysis if 2 eyes from one subject were included to account for a lack of independence between both eyes. A p-value of $\leq 0.05$ was considered statistically significant. Data was analyzed using R (R Core Team (2019). R: URL https://www.R-project.org/).

## Results

### Relationship between laser spot count and change in refractive error over time

We analyzed changes in refractive error over time in 153 eyes from 78 subjects (36 female, 42 male; 13 Hispanic, 24 black non-Hispanic, 32 white non-Hispanic, 9 other race/ethnicity; gestational age 25 ± 2 weeks; birth weight 737 ± 248 g) who underwent laser treatment for ROP between 2010 and 2019 by a single provider at a single center. 111 eyes were treated for ROP in zone II and 42 eyes were treated for ROP in zone III. No eyes were treated for ROP in Zone I. The mean post-menstrual age at treatment was 41 ± 8 weeks. The mean number of laser spots administered was 1793 ± 728 per eye. The subjects were followed for 37 ± 29 months with 3 ± 2 refractive error measurements during that period (Fig 1). Three eyes of 2 patients had macular dragging noted on exam.

Change in refractive error over time was chosen as the primary outcome rather than refractive error at a single time point as the best way to compare longitudinal data between eyes due to the heterogeneity of ages at which refractive error measurements were obtained. The mean refractive error appeared to change more between the first and second years of life than between the second and third years of life (Table 1), but statistical analysis of how the rate of change varied over time was not possible due to the heterogeneity of the data. Therefore, we utilized a linear model to make comparisons between eyes although some of the data may have been better fit by a bilinear or higher order model [13].

A multivariate linear mixed effects model was used to model change in refractive error over time. Birth weight, age at birth, age at treatment, plus, stage and zone were included as covariates in the multivariate model based on clinical relevance and the results of a bivariate analysis with laser spots as the primary predictor variable. The use of laser spots as a surrogate for the area of retina treated was supported by a very tight correlation between zone and laser spots (p <0.001). In the multivariate model, we found that eyes that received a greater number of laser spots had significantly greater change in refractive error over time. The mean rate of change in refractive error was -0.29 ± 0.48 D/year. The coefficient for the interaction between laser spots

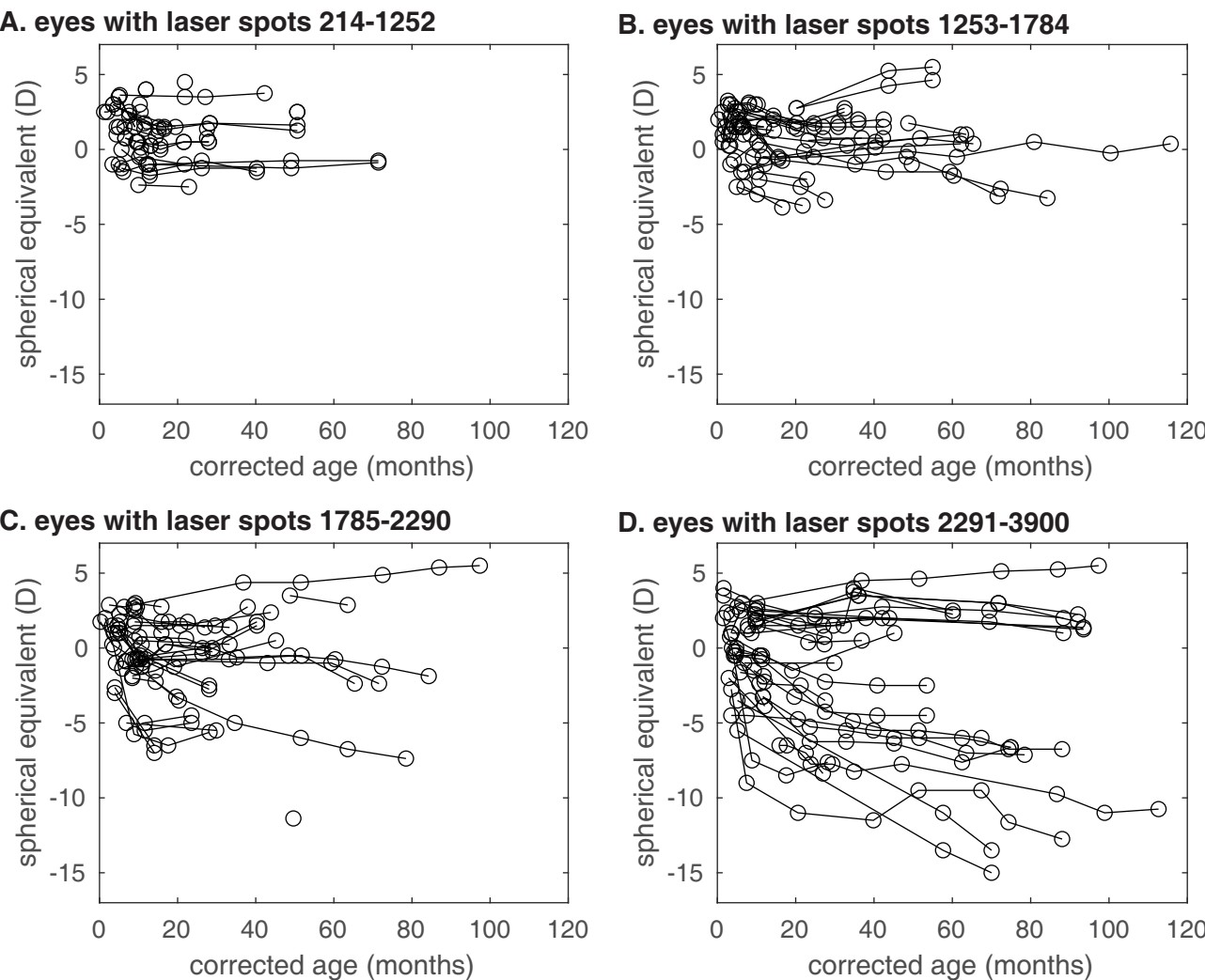

**Fig 1. Changes in refractive error with age.** One line represents one eye. Eyes were plotted in subgroups based on the number of laser spots received during treatment for retinopathy of prematurity. (A) first quartile of laser spots; (B) second quartile; (C) third quartile; (D) fourth quartile.

and time was -0.30 D (95% confidence interval -0.53 to -0.08), signifying that an increase in the number of laser spots by 1,000 is associated with additional myopia of 0.30 D per year. The coefficient in the bivariate model was similar (-0.31), indicating that other covariates are not confounding the association between spherical equivalent and the interaction of laser spots and time. Many eyes in the highest quartile in terms of number of laser spots (Fig 1D) did not develop myopia, indicating that there are other factors affecting myopia development after laser for ROP.

**Table 1. Mean refractive error in eyes treated with laser for retinopathy of prematurity.**

| Corrected Age | Spherical Equivalent ± SD | Number of Eyes |
|---|---|---|
| 0 - <1 year | +0.4 ± 2.3 D | 132 |
| 1 - <2 years | -1.3 ± 3.2 D | 70 |
| 2 - <3 years | -0.8 ± 3.1 D | 56 |

**Table 2. Early refraction compared to final refraction for eyes that underwent laser treatment for retinopathy of prematurity.**

| Early refraction | final refraction | number of eyes |
|---|---|---|
| hyperopia | hyperopia | 13 |
| hyperopia | emmetropia | 3 |
| hyperopia | myopia | 0 |
| emmetropia | hyperopia | 1 |
| emmetropia | emmetropia | 7 |
| emmetropia | myopia | 4 |
| myopia | hyperopia | 0 |
| myopia | emmetropia | 0 |
| myopia | myopia | 12 |

*early*, refraction at latest follow up prior to and including 18 months of age.

*final*, refraction at last follow up after 24 months of age.

*hyperopia*, spherical equivalent $\geq$ +1 D.

*emmetropia*, spherical equivalent $\geq$ - 1D and < +1D.

*myopia*, spherical equivalent < -1 D.

## Comparison of early and final refractive error

In the preceding analysis, we observed that much of the change in refractive error occurred in the first 1–2 years of life. We then hypothesized that refractive error before 18 months would be predictive of final refractive error. For the 40 subjects treated with laser who had refractive error measurements at age $\leq$ 18 months and > 24 months, early ($\leq$ 18 months) and final refractive error were compared (Table 2). For each subject, the eye with the largest magnitude of refractive error was selected for inclusion. 32 of eyes remained in the same refractive error category. None of the early hyperopic eyes became myopic, and all of the early myopic eyes remained myopic. However, 4 of the 12 early emmetropic eyes became myopic.

## Discussion

In subjects treated with laser for ROP, we found that more posterior disease and a larger area of treated retina corresponded to a greater degree of myopia over long term follow up. Since a single provider performed all of the laser treatments in a standard fashion, we were able to use the number of laser spots as a surrogate for the anterior-posterior location of disease and the area of treated retina. We hypothesized that eyes with more posterior disease would develop greater myopia. We found higher myopia in eyes that received a higher number of laser spots, which correlates well with findings of previous studies correlating laser spot count with myopia [8–11]. Previous studies analyzed refractive error at a single time point, and in contrast, we investigated change in refractive error from baseline and we found that myopia progressed more quickly in eyes that received a greater number of laser spots. Earlier studies included data from multiple treating physicians, and the number of laser spots could have reflected variations in treatment density between physicians rather than the area of retina treated. In our study, all subjects were treated by a single surgeon. Our interpretation of laser spot count as a surrogate for treated area is also supported by our finding of a high correlation between spots and zone, as well as the work of Young-Zvandasara et al. [9], who measured lasered area on photographs and correlated this with the number of laser spots administered.

The myopia of prematurity and of spontaneously regressed ROP usually increases in early life and remains fairly stable thereafter [3,4,7]. In our study of eyes that underwent laser for

ROP, we found that myopia developed primarily within the first two years of life, consistent with previous reports. Of importance to determining potential follow-up for children with a history of laser treatment for ROP, we found that in 32 of 40 eyes (80%), the refractive error category did not change between 18 months and the end of the follow up period (average of 37 months). The time course of development of myopia in ROP contrasts with school age myopia, which begins around ages 5–6 years and stabilizes in the teenage years. The follow up period of our study was not adequate to determine whether or not children treated with laser for ROP are at risk for school age myopia. The anatomic basis of myopia in prematurity and ROP differs from that of school age myopia as well. Premature eyes are characterized by greater lens thickness, shallower anterior chamber, and steeper corneas whereas in school age myopia, increased axial length is the primary abnormality [14–17]. Primate studies have indicated that defocus over the peripheral retina leads to local scleral changes in axial myopia, but does not alter anterior segment anatomy [18]. The role of the peripheral retina in myopia of ROP is suggested by data from studies including ours correlating a greater area of healthy peripheral retina with less myopia, but the mechanisms are likely to differ from that of axial myopia. Alternate experimental models that mimic the anterior segment changes seen in premature infants are needed.

An alternate measure of the area of avascular retina is zone of ROP at time of treatment. However, zone is a generalized categorical measure that encompasses a large area of retina and may be somewhat subjective, in contrast to laser spots, which are a more precise, continuous measure. Therefore, we chose to use the number of laser spots as our primary predicter variable. In the Early Treatment for Retinopathy of Prematurity study, they analyzed zone rather than laser spots, and found greater prevalence of high myopia in eyes treated for ROP in Zone I compared to Zone II [19]. Because of the lack of guidelines on when to treat ROP in Zone III, clinicians must weigh the benefits and risks. We included eyes treated for ROP in Zone III, and found that these eyes, which required fewer laser spots to treat, were less likely to develop myopia.

One limitation of analyzing change in refractive error over time is that clinically relevant differences in refractive error arising before the first measurement could be missed. The mean refractive error at years 0–1 of 0.4 D suggests that our first measurement was early enough to capture the development of myopia over time.

In conclusion, we found that myopia after ROP laser correlated with laser spot count when a consistent spot density was used. These findings support correlation between myopia and the area of avascular retina treated rather than spot density.

## Supporting information

**S1 Dataset. De-identified complete dataset.**
(XLSX)

## Author Contributions

**Conceptualization:** Iris S. Kassem, Joelle A. Hallak, Deborah M. Costakos.

**Data curation:** Eileen S. Hwang, Iris S. Kassem.

**Formal analysis:** Eileen S. Hwang, Rawan Allozi, Sasha Kravets, Joelle A. Hallak.

**Investigation:** Eileen S. Hwang, Iris S. Kassem, Khalid Y. Al-Kirwi, Deborah M. Costakos.

**Methodology:** Eileen S. Hwang, Iris S. Kassem, Rawan Allozi, Sasha Kravets, Joelle A. Hallak, Deborah M. Costakos.

**Project administration:** Eileen S. Hwang, Iris S. Kassem.

**Resources:** Iris S. Kassem.

**Supervision:** Eileen S. Hwang, Iris S. Kassem, Joelle A. Hallak, Deborah M. Costakos.

**Validation:** Eileen S. Hwang, Rawan Allozi, Joelle A. Hallak.

**Visualization:** Sasha Kravets.

**Writing – original draft:** Eileen S. Hwang, Iris S. Kassem.

**Writing – review & editing:** Eileen S. Hwang, Iris S. Kassem, Rawan Allozi, Sasha Kravets, Khalid Y. Al-Kirwi, Joelle A. Hallak, Deborah M. Costakos.

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
