## [Decision Letter · Decision Letter 0]

16 Aug 2022

PONE-D-22-09529

Association between myopia progression and quantity of laser treatment for retinopathy of prematurity

PLOS ONE

Dear Dr. Thomas R. Shearer,

Thank you for submitting your manuscript to PLOS ONE. After careful consideration, we feel that it has merit but does not fully meet PLOS ONE’s publication criteria as it currently stands. Therefore, we invite you to submit a revised version of the manuscript that addresses the points raised during the review process.

We look forward to receiving your revised manuscript.

Kind regards,

Shiying Li, MBBS

Academic Editor

PLOS ONE

https://journals.plos.org/plosone/s/file?id=ba62/PLOSOne_formatting_sample_title_authors_affiliations.pdf".

“I have read the journal's policy and the authors of this manuscript have the following competing interests:

Eileen S. Hwang received gifts, meals, travel support and/or education from Regeneron, Allergan, Valeant, Alimera Sciences, Alcon, Bausch & Lomb, Beaver Vistec International, Spark Therapeutics, and Katalys.

Iris S. Kassem's spouse is employed by AbbVie.

Rawan Allozsi was employed by AbbVie prior to commencing work on this project and became employed at The Janssen Pharmaceutical Companies of Johnson & Johnson after completing work on this project.”

Additional Editor Comments:

The topic of this study is interesting. Please see reviewers' comments and revise the MS accordingly.

Reviewers' comments:

Reviewer's Responses to Questions

**Comments to the Author**

1. Is the manuscript technically sound, and do the data support the conclusions?

Reviewer #1: Partly

Reviewer #2: Yes

2. Has the statistical analysis been performed appropriately and rigorously? 

Reviewer #1: I Don't Know

Reviewer #2: Yes

3. Have the authors made all data underlying the findings in their manuscript fully available?

Reviewer #1: Yes

Reviewer #2: Yes

4. Is the manuscript presented in an intelligible fashion and written in standard English?

Reviewer #1: Yes

Reviewer #2: Yes

5. Review Comments to the Author

Reviewer #1: The topic of this article has certain significance,which explore the impact of laser treatment on the visual develop in ROP childen. after carefuly reading the article,the following factors should be considered.

1. currently, Anti-VEGF has been partially replaced the laser treatment on ROP.so ,the advancement of this article were be weakend.

Myopia is a multifactorial disease, which is related to genetic, environmental and other factors. Therefore, the refractive status of the parents of the children with ROP should also be taken into account in the study.

3. Eye axis related to the aging and mopia, If the length of the ocular axial involved in the research, the conclusions are more credible.

Reviewer #2: Hwang et al. retrospectively investigated myopia development in laser-treated ROP. The authors report a statistical correlation between the number of applied laser spots and the degree of myopia. They conclude that higher myopia develops in infants that received more laser treatment. This study is interesting and clinically relevant. The following issues should be addressed before consideration for publication.

Major comments

1. CRYO-ROP demonstrated a strong correlation between severity of ROP and development of myopia, independent of treatment (Quinn 1992). The more posterior the disease, the higher was the risk of myopia development. As mentioned on page 8, there is similar data from ETROP. The more severe and the more posterior the disease, however, the more likely it also needs treatment.

This means that laser treatment (or the extent thereof) may just be a confounding factor in this scenario: Myopia development correlates with ROP severity, and only because more severe/more posterior disease requires more laser treatment, there appears to be a relationship between those latter two.

The only way to proof or disproof that retinal ablative treatment in ROP is indeed influencing myopia development is a randomized trial design. CRYO-ROP randomized one eye per infant to cryotherapy and the other to observation and did not find a statistical difference in subsequent myopia development in pairwise comparison (Quinn 2001). This provides strong evidence that myopia in ROP develops independent of retinal ablative treatment.

Today, clinicians in ROP treatment are faced with the challenging decision between laser and anti-VEGF. Laser has several advantages over anti-VEGF including dramatically shorter follow-up time, no risk of systemic side effects, and no endophthalmitis. To facilitate informed decision making, the non-evidence-based impression that laser causes myopia in ROP should be avoided.

These issues should be incorporated into the manuscript’s abstract and discussion. When stating that more laser is associated with more myopia, it should be clarified that more laser is a surrogate marker for more posterior/more severe disease.

Minor comments

2. According to US and international guidelines, zone III ROP does not normally require treatment. What was the rational for treating 42 eyes with zone III ROP in this study?

3. Page 3, line 53: Myopia secondary to ROP is usually not due to increased axial length and thus not resulting in myopic macular degeneration. Therefore, what is meant by “retinal disease” caused by myopia secondary to ROP? The cited reference #3 does not seem to provide supporting evidence or clarification for this.

6. PLOS authors have the option to publish the peer review history of their article (what does this mean?). If published, this will include your full peer review and any attached files.

Reviewer #1: No

Reviewer #2: No

---

## [Author Response · Author response to Decision Letter 0]

27 Sep 2022

Reviewer #1: The topic of this article has certain significance,which explore the impact of laser treatment on the visual develop in ROP childen. after carefuly reading the article,the following factors should be considered.

1. currently, Anti-VEGF has been partially replaced the laser treatment on ROP.so ,the advancement of this article were be weakend.

Our finding of less myopia after treatment for anterior ROP suggests that laser may be favored over anti-VEGF if the disease is far anterior.

2. Myopia is a multifactorial disease, which is related to genetic, environmental and other factors. Therefore, the refractive status of the parents of the children with ROP should also be taken into account in the study.

We agree that a confounder could be the refractive status of the parents. Unfortunately, this information was not available in this retrospective study. However, we are not aware of any reason to believe that parental refractive error correlates with zone of ROP so we do not think this affected our final conclusions.

3. Eye axis related to the aging and mopia, If the length of the ocular axial involved in the research, the conclusions are more credible.

Post-ROP myopia is not axial myopia (see references below). Unfortunately, this information was not collected in this retrospective study.

15. Wu WC, Lin RI, Shih CP, Wang NK, Chen YP, Chao AN, et al. Visual acuity, optical components, and macular abnormalities in patients with a history of retinopathy of prematurity. Ophthalmology. 2012;119(9):1907–16. 

16. Cook A, White S, Batterbury M, Clark D. Ocular growth and refractive error development in premature infants with or without retinopathy of prematurity. Investig Ophthalmol Vis Sci. 2008;49(12):5199–207. 

Reviewer #2: Hwang et al. retrospectively investigated myopia development in laser-treated ROP. The authors report a statistical correlation between the number of applied laser spots and the degree of myopia. They conclude that higher myopia develops in infants that received more laser treatment. This study is interesting and clinically relevant. The following issues should be addressed before consideration for publication.

Major comments

1. CRYO-ROP demonstrated a strong correlation between severity of ROP and development of myopia, independent of treatment (Quinn 1992). The more posterior the disease, the higher was the risk of myopia development. As mentioned on page 8, there is similar data from ETROP. The more severe and the more posterior the disease, however, the more likely it also needs treatment.

This means that laser treatment (or the extent thereof) may just be a confounding factor in this scenario: Myopia development correlates with ROP severity, and only because more severe/more posterior disease requires more laser treatment, there appears to be a relationship between those latter two.

The only way to proof or disproof that retinal ablative treatment in ROP is indeed influencing myopia development is a randomized trial design. CRYO-ROP randomized one eye per infant to cryotherapy and the other to observation and did not find a statistical difference in subsequent myopia development in pairwise comparison (Quinn 2001). This provides strong evidence that myopia in ROP develops independent of retinal ablative treatment.

Today, clinicians in ROP treatment are faced with the challenging decision between laser and anti-VEGF. Laser has several advantages over anti-VEGF including dramatically shorter follow-up time, no risk of systemic side effects, and no endophthalmitis. To facilitate informed decision making, the non-evidence-based impression that laser causes myopia in ROP should be avoided.

These issues should be incorporated into the manuscript’s abstract and discussion. When stating that more laser is associated with more myopia, it should be clarified that more laser is a surrogate marker for more posterior/more severe disease.

We appreciate this reviewer's perspective that the greater myopia may be due to more posterior location of active ROP disease. We incorporated this concept into the abstract, introduction and discussion.

The background section of the abstract now reads:

Previous studies found that infants with retinopathy of prematurity (ROP) who were treated for more posterior disease with a greater number of laser spots developed higher myopia. These studies included multiple physicians with variations in laser density. In treatments by a single physician, laser spot count is a better surrogate for area of avascular retina and anterior-posterior location of disease, so that the relationship with myopia can be better assessed.

The second paragraph of the introduction now reads:

Myopia occurs after ROP treatment, and the risk of myopia is also elevated in spontaneously regressed ROP [3,4]. The risk of myopia is higher after spontaneous regression of posterior disease (i.e. zone 2) compared to anterior disease (i.e. zone 3) [5]. The CRYO-ROP study found more high myopia in treated compared to control eyes, although there were more eyes that could not be refracted in the control group [6]. In the subgroup of subjects that could be refracted in both eyes, there was no difference in myopia rates between treated and control eyes [6]. Anterior-posterior location of active ROP disease correlates with myopia, as well as the extent of final vascularization after spontaneous regression [7]. During laser treatment for ROP, the entire area of avascular retina is ablated to reduce the drive for neovascularization. Posterior location of ROP, and thereby a larger area of avascular retina ablated during laser treatment, may correlate with a greater risk of myopia [8–12]. Precisely measuring the anterior-posterior location or the area of retina treated are difficult.

The first paragraph of the discussion now reads:

In subjects treated with laser for ROP, we found that more posterior disease and a larger area of treated retina corresponded to a greater degree of myopia over long term follow up. Since a single provider performed all of the laser treatments in a standard fashion, we were able to use the number of laser spots as a surrogate for the anterior-posterior location of disease and the area of treated retina. We hypothesized that eyes with more posterior disease would develop greater myopia.

Minor comments

2. According to US and international guidelines, zone III ROP does not normally require treatment. What was the rational for treating 42 eyes with zone III ROP in this study?

Since there are no guidelines regarding whether to treat or observe ROP in zone III, practice patterns vary. Zone III subjects were treated to reduce the need for intensive follow up with scleral depressed exams. Our inclusion of treatments for ROP in zone III actually strengthen our evaluation of the correlation between spot counts and myopia by providing greater variation in spot counts, since laser in zone III is to a smaller area that requires fewer laser spots. Our data provide evidence that treatment in zone III is unlikely to create myopia.

3. Page 3, line 53: Myopia secondary to ROP is usually not due to increased axial length and thus not resulting in myopic macular degeneration. Therefore, what is meant by “retinal disease” caused by myopia secondary to ROP? The cited reference #3 does not seem to provide supporting evidence or clarification for this.

This sentence was removed.

---

## [Decision Letter · Decision Letter 1]

6 Dec 2022

PONE-D-22-09529R1Association between myopia progression and quantity of laser treatment for retinopathy of prematurityPLOS ONE

Dear Dr. Hwang,

Thank you for submitting your manuscript to PLOS ONE. After careful consideration, we feel that it has merit but does not fully meet PLOS ONE’s publication criteria as it currently stands. Therefore, we invite you to submit a revised version of the manuscript that addresses the points raised during the review process

We look forward to receiving your revised manuscript.

Kind regards,

Shiying Li, MBBS

Academic Editor

PLOS ONE

Journal Requirements:

Additional Editor Comments :

Please see reviewers' comments and revise the MS accordingly before we accept it.

Reviewers' comments:

Reviewer's Responses to Questions

**Comments to the Author**

1. If the authors have adequately addressed your comments raised in a previous round of review and you feel that this manuscript is now acceptable for publication, you may indicate that here to bypass the “Comments to the Author” section, enter your conflict of interest statement in the “Confidential to Editor” section, and submit your "Accept" recommendation.

Reviewer #2: All comments have been addressed

Reviewer #3: (No Response)

2. Is the manuscript technically sound, and do the data support the conclusions?

Reviewer #2: (No Response)

Reviewer #3: Yes

3. Has the statistical analysis been performed appropriately and rigorously? 

Reviewer #2: (No Response)

Reviewer #3: Yes

4. Have the authors made all data underlying the findings in their manuscript fully available?

Reviewer #2: (No Response)

Reviewer #3: Yes

5. Is the manuscript presented in an intelligible fashion and written in standard English?

Reviewer #2: (No Response)

Reviewer #3: Yes

6. Review Comments to the Author

Reviewer #2: (No Response)

Reviewer #3: The authors retrospective studied the ROP-treated infants and found the correlation between myopia and the area of avascular retina treated rather than spot density. However, there are some questions as below.

1. Line 50~53, please add some references for this description.

2. Line 123, please keep only one ‘.’ In the end of ‘variable’

3. Line 149~152, as you described, ‘32 of eyes…’, you’d better discuss these results in the discussion.

4. In ‘Discussion’, it would more interesting if you discussed some mechanisms of the area of avascular retina contributed to the myopia beyond your hypothesis in line 163~165.

7. PLOS authors have the option to publish the peer review history of their article (what does this mean?). If published, this will include your full peer review and any attached files.

Reviewer #2: No

Reviewer #3: No

---

## [Author Response · Author response to Decision Letter 1]

11 Dec 2022

Journal Requirements:

Response: We reviewed our references list and found it to be complete and correct. We have not included any retracted papers.

Additional Editor Comments :

Please see reviewers' comments and revise the MS accordingly before we accept it.

We have revised the manuscript according to the reviewers' comments.

Reviewer #3: The authors retrospective studied the ROP-treated infants and found the correlation between myopia and the area of avascular retina treated rather than spot density. However, there are some questions as below.

1. Line 50~53, please add some references for this description.

Response: added an additional reference in line 51 to:

Dikopf MS, Machen LA, Hallak JA, Chau FY, Kassem IS. Zone of retinal vascularization and refractive error in premature eyes with and without spontaneously regressed retinopathy of prematurity. Journal of AAPOS. 2019;23(4):211.e1-6. 

2. Line 123, please keep only one ‘.’ In the end of ‘variable’

Response: the double period has been corrected to a single period at the end of line 128.

3. Line 149~152, as you described, ‘32 of eyes…’, you’d better discuss these results in the discussion.

Response: We added a sentence to the discussion, lines 177-178, "we found that in 32 of 40 eyes (80%), the refractive error category did not change between 18 months and the end of the follow up period (average of 37 months)."

4. In ‘Discussion’, it would more interesting if you discussed some mechanisms of the area of avascular retina contributed to the myopia beyond your hypothesis in line 163~165.

Response: I edited the discussion (lines 185-191) to read: Primate studies have indicated that defocus over the peripheral retina leads to local scleral changes in axial myopia, but does not alter anterior segment anatomy [18]. The role of the peripheral retina in myopia of ROP is suggested by data from studies including ours correlating a greater area of healthy peripheral retina with less myopia, but the mechanisms are likely to differ from that of axial myopia. Alternate experimental models that mimic the anterior segment changes seen in premature infants are needed.

---

## [Editor Report · Decision Letter 2]

19 Dec 2022

Association between myopia progression and quantity of laser treatment for retinopathy of prematurity

PONE-D-22-09529R2

Dear Dr. Hwang,

We’re pleased to inform you that your manuscript has been judged scientifically suitable for publication and will be formally accepted for publication once it meets all outstanding technical requirements.

Kind regards,

Shiying Li, MBBS

Academic Editor

PLOS ONE

Additional Editor Comments (optional):

All the comments are responded.
---

## [Editor Report · Acceptance letter]

21 Dec 2022

PONE-D-22-09529R2 

Association between myopia progression and quantity of laser treatment for retinopathy of prematurity 

Dear Dr. Hwang:

I'm pleased to inform you that your manuscript has been deemed suitable for publication in PLOS ONE. Congratulations! Your manuscript is now with our production department. 

Kind regards, 

on behalf of

Dr. Shiying Li 

Academic Editor

PLOS ONE